# Increased Vocalization of Rats in Response to Ultrasonic Playback as a Sign of Hypervigilance Following Fear Conditioning

**DOI:** 10.3390/brainsci11080970

**Published:** 2021-07-23

**Authors:** Krzysztof H. Olszyński, Rafał Polowy, Agnieszka D. Wardak, Aneta W. Grymanowska, Robert K. Filipkowski

**Affiliations:** Behavior and Metabolism Research Laboratory, Mossakowski Medical Research Institute, Polish Academy of Sciences, 5 Pawinskiego St, PL02-106 Warsaw, Poland; kholszynski@imdik.pan.pl (K.H.O.); rpolowy@imdik.pan.pl (R.P.); awardak@imdik.pan.pl (A.D.W.); agrymanowska@imdik.pan.pl (A.W.G.)

**Keywords:** hypervigilance, hyperreactivity, exaggerated reactivity, generalization, PTSD, anxiety, depression, heart rate, ultrasonic vocalizations, Wistar

## Abstract

We investigated the effects of prior stress on rats’ responses to 50-kHz (appetitive) and 22-kHz (aversive) ultrasonic playback. Rats were treated with 0, 1, 6 or 10 shocks (1 s, 1.0 mA each) and were exposed to playbacks the following day. Previous findings were confirmed: (i) rats moved faster during 50-kHz playback and slowed down after 22-kHz playback; (ii) they all approached the speaker, which was more pronounced during and following 50-kHz playback than 22-kHz playback; (iii) 50-kHz playback caused heart rate (HR) increase; 22-kHz playback caused HR decrease; (iv) the rats vocalized more often during and following 50-kHz playback than 22-kHz playback. The previous shock affected the rats such that singly-shocked rats showed lower HR throughout the experiment and a smaller HR response to 50-kHz playback compared to controls and other shocked groups. Interestingly, all pre-shocked rats showed higher locomotor activity during 50-kHz playback and a more significant decrease in activity following 22-kHz playback; they vocalized more often, their ultrasonic vocalizations (USV) were longer and at a higher frequency than those of the control animals. These last two observations could point to hypervigilance, a symptom of post-traumatic stress disorder (PTSD) in human patients. Increased vocalization may be a valuable measure of hypervigilance used for PTSD modeling.

## 1. Introduction

Prior stress and its effects on rodents’ subsequent behavior have been extensively studied and reflect human symptoms of neuropsychiatric disorders. For example, foot shocks of varying intensity produce behavioral and neurochemical changes which model depression, anxiety, and post-traumatic stress disorder (PTSD) [1,2]. Specifically, electric shocks delivered during fear conditioning in both learning- and trauma-imitating protocols (compared in [3]) were shown to result in increased avoidance, cognitive and mood alterations, increased arousal, social avoidance and sleep disturbance (summarized in [4])

Conditioned fear in rodents is measured by freezing levels to cues or contexts previously paired with the shock. Conditioned fear may be observed in different or partially altered contexts due to fear generalization [5,6,7,8]. However, acute exposure to non-traumatic stress (e.g., sound and light) increased locomotor activity, while chronic stress reduced basal activity and eliminated the activation response to acute stress [9].

Heart rate (HR) is one of the most studied psychophysiological parameters in anxiety disorders. It was demonstrated that successful psychological treatment reduces HR physiological reactivity in patients with PTSD and possibly other anxiety disorders [10,11]. However, results concerning HR changes in fear conditioning paradigms are equivocal. Nijsen et al. reported less pronounced tachycardia in the cage where rats experienced fear conditioning compared with control rats [12], while Carnvali et al. showed a reduction of HR lasting for five days after fear conditioning [13]. However, re-exposure of rats to the context where they had received electrical foot shocks evoked an immediate increase in HR [14,15] or a delayed rise in HR [16,17], also in a latent inhibition protocol [18]. Additionally, a decrease in the daily HR rhythm amplitude was observed in a social defeat protocol [19].

Rats communicate via several sensory channels including the emission of ultrasonic vocalizations (USV). USV of 22-kHz type signal aversive states, while 50-kHz USV signal appetitive states [20,21,22,23]. Rats emitted 22-kHz USV during fear conditioning and re-exposure to the test apparatus, which corresponded with the number of shocks received [24]. Aversive and appetitive USV emitted by rats or played from speakers evoke physiological and emotional changes in conspecifics [25].

We recently discovered changes in locomotion, USV emission, and HR in Wistar rats exposed to ultrasonic playback from a speaker [26]. In particular, 50-kHz playback elicited abundant appetitive vocalization, increased HR, and locomotor activity. In contrast, 22-kHz playback led to an abrupt decrease in HR and locomotor activity. We wanted to establish whether prior stress affects rats’ reactions to ultrasonic playback and whether the response is dependent on the number of shocks previously received (dose-dependent). Towards this end, we fear-conditioned (FC) rats with varying numbers of shocks, encompassing the usually used range [3], and investigated changes in locomotion, USV emission and HR in rats exposed to 50-kHz vs. 22-kHz USV playback.

## 2. Materials and Methods

### 2.1. Animals and Housing

All experiments were approved by the Second Local Ethical Committee in Warsaw. Naïve adult male Wistar rats (7 weeks of age, from The Center for Experimental Medicine of the Medical University of Bialystok, Poland), control animals (no electric shock, 0-Trial), and conditioned rats (receiving 1, 6 or 10 conditioning stimuli; 1-Trial, 6-Trial and 10-Trial, respectively) were kept in pairs in two separate rooms. Standard chow and water were provided ad libitum. Both housing rooms had a 12 h light–dark cycle and an ambient temperature of 22–25 °C. Fear conditioning was conducted between 15:00 and 24:00 h when the overall noise level in the animal house is low. All playback experiments were conducted during the light cycle (9:00–21:00 h) on the weekend. There were four weeks between rat arrival and the start of the experiment. In the first week, the rats were habituated to new facilities. All animals were handled once for 2 min per day for 12 days before the experiment. The three experimenters each had at least four handling sessions with the rats. Surgery was performed on the third week.

### 2.2. Surgical Procedures

A radiotelemetric transmitter (HD-S10, Data Sciences International, St. Paul, MN, USA) was implanted into the abdominal aorta as previously described [26]. The radiotelemetric transmitter (HD-S10, Data Sciences International, St. Paul, MN, USA) for cardiovascular studies was disinfected using Cidex^®^ (Johnson&Johnson, New Brunswick, NJ, USA) and was implanted under ketamine-xylazine anesthesia. The abdominal region was shaved and disinfected (Octenisept, Schulke, Norderstedt, Germany). A midline incision was performed, and the transmitter sensor was implanted into the abdominal aorta by direct puncturing of the vessel (21 g needle) and fixed with tissue glue (Histoacryl^®^, B. Braun, Melsungen, Germany). The transmitter body was placed in the peritoneal cavity and fixed to the abdominal muscle wall. After surgery, the animal was subcutaneously injected with Metacam (0.4 mg/kg; Boehringer Ingelheim, Ingelheim am Rhein, Germany) for analgesia. An illustrative image with the surgery details can be found elsewhere; Figure 5 in [27]; please note, tissue glue was used instead of cellulose patches and silk sutures. Rats were given at least seven days post-surgery for recovery before the start of the experiment. During recovery, the animals were handled and habituated to the conditions of playback experiment four times. The animals were all 12 weeks of age at the start of the experiment.

### 2.3. Fear Conditioning

The animals were transported individually to the fear-conditioning room and placed in a sound-attenuated fear conditioning apparatus (MED-VFC2-USB-R, Med Associates, Fairfax, VT, USA; with insides of 54.64 × 64.04 × 29.21 cm). Each animal was habituated to the cage for 10 min with no light inside; rats’ freezing, defined as the absence of movement for at least 1 s, was scored automatically by Med Associates Video Freeze software during the first 5 min. The cage was cleaned between animals using detergent, wiped using 10% ethanol and was allowed to dry. The next day, rats were placed in the conditioning cage with no light inside. After 5-min habituation, they received 1, 6 or 10 conditioning stimuli which consisted of a 20 s long white light co-terminating with an electric foot-shock (1 s, 1 mA). The inter-trial interval (ITI) ranged from 180 to 300 s (mean, 240 s) (comp. [28]). Therefore, the conditioning procedure differed in length between groups: 9 min 20 s for 1-Trial, 31 min for 6-Trial, and 48 min 20 s for 10-Trial groups. An equal-time-length control group (no shock) was used for each group. A playback experiment was given one day later (see the next paragraph); the following day (two days after conditioning), rats were returned to the same fear-conditioning context to measure freezing levels (Test). After 5-min habituation, rats were exposed to three blocks of 20 s of white light (CS) followed by 5 min of silence. Freezing was evaluated during the habituation and exposure to CS. The conditioning procedure was executed by an investigator not involved in the playback experiment.

### 2.4. Playback Experiment

One day after the conditioning, the rats were transferred into individual experimental cages, identical to home cages (plastic; 37 × 25 × 16 cm), and transported to the experimental room, where under white light, in the absence of the experimenter and other rats in the room, acoustic stimuli were presented through an ultrasonic speaker (Vifa, Avisoft Bioacoustics, Berlin, Germany), placed just above the shorter side of the cage, connected to an UltraSoundGate Player 116 (Avisoft Bioacoustics). USV emitted by the rat were recorded by a CM16/CMPA condenser microphone (UltraSoundGate, Avisoft Bioacoustics) placed 33 cm above the center of the cage floor, 20 cm away from the speaker. In this configuration, calls from the speaker were still visible in the recording (monitoring of playback), but they were distinctively weaker than USV emitted from the cage. Both playback and recording were performed using Avisoft Recorder USGH software (Avisoft Bioacoustics). The locomotor activity of the animal was recorded with a camera (acA1300-60gc, Basler AG, Ahrensburg, Germany) mounted above the cage and EthoVision XT software (version 10, Noldus, Wageningen, The Netherlands). Signals from radiotelemetric transmitters were collected by receivers located under the cage floor and then recorded by Ponemah software (version 6.32, Data Sciences International, St. Paul, MN, USA).

### 2.5. Ultrasonic Playback Presentation

Upon placing a rat into the experimental cage, 10 min of silence with a turned-on speaker, that is, background noise of 20.6 ± 0.2 dB, was followed by four 10-s-long sets of signals, separated by 5-min-long silence intervals; see Figure 1 in [26]. Four sets of signals (playbacks) were presented in counterbalanced order to each rat: (i) 50-kHz natural calls (referred to as “50-kHz USV”), 84 calls in 3 repeats, of 49.2 to 73.4 kHz frequency and 58.6 ± 0.7 kHz mean peak frequency, 28.4 ± 1.6 ms duration, 31.9 ± 0.6 dB sound pressure, recorded during rats’ social interactions; (ii) 50-kHz software-generated tones (“50-kHz tones”), 32.6 ± 0.7 dB; (iii) 22-kHz natural calls (“22-kHz USV”), 24 calls in 8 repeats, 21.4–23.0 kHz, 22.1 ± 0.1 kHz, 375.3 ± 21.6 ms, 38.3 ± 1.2 dB, recorded during fear conditioning (Avisoft Bioacoustics [Internet]; c2020, Examples of rat ultrasonic vocalizations (USV), Norwegian rat (*Rattus norvegicus*), Wistar albino strain, males); and (iv) 22-kHz software-generated tones (“22-kHz tones”), 43.3 ± 3.0 dB, although the sound playbacks of the same frequency range, for example, 50-kHz USV and 50-kHz tones, always followed each other. Artificial tones were generated based on natural tones (mean peak frequency, duration, pauses between tones in the set, but with no frequency modulations) using Avisoft SASLab Pro (Avisoft Bioacoustic). Calls were presented with a sampling rate of 200 kHz in 16-bit format. The sound pressure levels of the background noise and playback signals were assessed in the middle of the test cage’s floor, at the height of the animals’ typical head position, facing the speaker.

### 2.6. Analysis of USV and Locomotor Activity

Recordings were transferred to SASLab Pro (Avisoft Bioacoustics), and a fast Fourier transform was conducted (512 FFT-length, 100% frame, Hamming window and 75% time window overlap), resulting in high resolution spectrograms (frequency resolution: 391 Hz; time resolution: 0.64 ms). USV recordings were analyzed using SASLab Pro 5.2.xx. Spectrograms were generated from the.wav files with the following parameters: window type: FlatTop, 512 FFT length, 100% frame size and 75% temporal resolution overlap. An experienced user scored USV on the spectrogram. For analysis, mean peak frequency and element duration were taken via values measured by the software. Automated video tracking system (Ethovision, Noldus, Wageningen, The Netherlands) was used to measure the total distance traveled (cm), a measure of general locomotor activity, and proximity to speaker, that is, time spent (%) in the half of the cage closer to the speaker. The center-point of each animal’s shape was used as a reference point for measurements of locomotor activity, thus registering only full-body movements, that is, distance traveled by a given rat.

### 2.7. Statistical Analysis

All data were analyzed using non-parametric Friedman, Wilcoxon, and Mann-Whitney tests with Statistica 7.1 (Stat-Soft); the *p* values are given, with a minimal level of significance of *p* < 0.05. Figures were prepared using GraphPad Prism 7 software and depict average values with a standard error of the mean (SEM). Based on preconditioning evaluation (Appendix A), two rats from the 6-Trial group were excluded as outliers (i.e., emitting exceptionally many USV, >3 × Standard Deviation) and subsequently removed from the analysis. However, every reported significant *p* value was verified to be present with the two rats included (apart from a few exceptions within the supplementary tables marked appropriately).

## 3. Results

### 3.1. Rats Showed Freezing after Fear Conditioning

Before conditioning, the rats from all four groups (0, 1, 6, 10-Trial) emitted similar numbers of USV (Appendix A) and showed similar HR (Appendix A) and freezing levels (Appendix A). During the test (two days after conditioning), that is, one day after the playback experiment, the rats from 1-Trial, 6-Trial, and 10-Trial groups showed increased freezing (Appendix A).

### 3.2. Except for the Periods of Ultrasonic Playback, Rats’ Behavior Remained Relatively Constant

Locomotor activity, measured as distance traveled (please note, graphs contain distances travelled in cm per 10 s; speed, when mentioned, is reflected by the distances divided by 10, that is, in cm/s), was the same during the 10-min-silence period at an average speed of 1.70 cm/s (Appendix A), which declined during the playback session to 1.30 cm/s (*p* = 0.0000, Figure 1), for example, to 1.31 cm/s (*p* = 0.0000) and to 1.14 cm/s (*p* = 0.0000) during our control time-intervals (comp. [26], that is, from −120 s to −100 s and −30 s to −10 s, respectively, all Wilcoxon). Within these periods, however, the distance travelled remained relatively constant (Figure 1, Figure 2 and Figure 3, Appendix A). Please note that Figure 3 has a guiding explanation regarding figures content and some take-home messages.

We did not observe a strong preference for either side of the cage during the initial 10 min (Appendix A). Importantly, side-preference was not observed in the playback session, during either of the two control time intervals, before any of the four kinds of playbacks, or in the five groups (including the all-rats group) (Appendix A; *p* > 0.05 in all 40 cases, all Wilcoxon). The rats had no cage-side preference before playback presentation, as noted by values around 50% (dotted line) before each ultrasonic playback (Figure 1, Figure 2 and Figure 3, Appendix A), as well as a relative lack of changes in preference within the control intervals (Appendix A).

### 3.3. Animals Moved Faster during 50-kHz Ultrasonic Presentations and Slowed Down after 22-kHz Ultrasonic Presentations

All rats traveled significantly longer distances during the presentation of 50-kHz signals (Figure 1, Figure 2 and Appendix A; note the *p* values for −10–10 s time-intervals in Appendix A), that is, at 0 s time-interval vs. neighboring −10 s and 10 s time-intervals. These pair-comparisons were significant for both USV and tone playbacks in all rats (*p* = 0.0000 in all four cases, Wilcoxon). The differences were also significant in all four groups, for 50-kHz USV playback (all eight cases, Wilcoxon, with 0.0000–0.0026 *p* levels).

In the case of 22-kHz playback (Figure 1, Figure 2 and Appendix A; note the *p* values for 0–30 s time-intervals in Appendix A), a reduction in distance traveled appeared immediately after signal presentation, that is, at 0 vs. 10 s time-intervals after both USV and tone playbacks in all rats (*p* = 0.0000 for both conditions, Wilcoxon). These differences were also significant in all groups, for 22-kHz USV playback (all four cases, Wilcoxon, with 0.0003–0.0438 *p* levels).

### 3.4. Conditioned Rats Showed Higher Locomotor Activity during 50-kHz Playback

The peak of locomotor activity observed during 50-kHz playback seemed to be higher in FC rats (Figure 1, Figure 2 and Appendix A). Indeed, when distances traveled during 50-kHz USV and tone playbacks, that is, at 0 s time-interval, were averaged for analysis, the group comparisons revealed the difference between the 0-Trial rats (2.51 ± 0.20 cm/s) and all fear-conditioned rats combined (3.24 ± 0.19 cm/s, *p* = 0.0247); this was observed for 50-kHz USV playback as well, that is, between 0-Trial rats (2.62 ± 0.24 cm/s) and all fear-conditioned rats combined (3.45 ± 0.22 cm/s, *p* = 0.0125/.0095), as well as 6-Trial and 10-Trial rats combined (3.54 ± 0.29 cm/s, *p* = 0.0210, all Mann-Whitney; Figure 2A,C).

### 3.5. Conditioned Rats Showed a More Significant Decrease in Activity Following 22-kHz Playback

In the case of 22-kHz playback (Figure 1, Figure 2 and Appendix A), a reduction in distance traveled was observed immediately after signal presentation, that is, the difference in locomotor activity at 10 s vs. 0 s time-intervals appeared to be higher in conditioned animals, for example, for 22-kHz USV playback, it was 1.02 ± 0.21 cm/s vs. 0.39 ± 0.21 cm/s in the control group (*p* = 0.0525).

This effect was more pronounced when data following USV and tone playbacks were averaged for analysis; with a 0.36 ± 0.12 cm/s reduction in 0-Trial vs. a 0.82 ± 0.21 cm/s reduction in 6-Trial and 10-Trial rats combined (*p* = 0.0188), and vs. a 0.84 ± 0.17 cm/s reduction in all fear-conditioned rats (*p* = 0.0175; Figure 2E,G).

### 3.6. All Rats Approached the Speaker; It Was More Pronounced during and Following 50-kHz Playback

Both during and following playbacks, the rats from all groups approached the speaker (Figure 1, Figure 2 and Figure 3, Appendix A). Interestingly, it was observed for both 50 and 22-kHz playbacks; note the *p* values for 10–30 and 10–60 s time-intervals in Appendix A compared with 50% chance levels as well as at all time intervals, which included 0 s time-interval vs. control intervals in Appendix A. An increase in time spent in the speaker’s half of the cage between before vs. after/during the playbacks was observed in all groups (Appendix A).

However, the rats spent more time in the speaker’s half of the cage when presented with 50-kHz playback than when exposed to 22-kHz sounds (Figure 1A,C,E,G). For USV playback, the difference was significant during playback (*p* = 0.0477) and during the 10–30 s time-interval (*p* = 0.0088), while for the tone playback, the difference was significant only during 10–30 s time-interval (*p* = 0.0145). When results from USV and tone playbacks were averaged for analysis, the difference was even more significant during playback (*p* = 0.0041) and the 10–30 s time-interval (*p* = 0.0003, all Wilcoxon). There was no significant difference in control time-intervals for these comparisons, that is, from −120 s to −100 s and −30 s to −10 s (*p* > 0.05, Wilcoxon).

### 3.7. HR Levels Declined during the Whole Experimental Session

When average levels of HR from the first 5 min of the 10-min-silence period were compared with those from the last 5 min of the playback session, there was a significant decline in the HR of the 0-Trial (476.6 ± 5.5 vs. 383.2 ± 4.9, *p* = 0.0000), 1-Trial (461.0 ± 7.3 vs. 367.8 ± 6.2, *p* = 0.0005), 6-Trial (482.7 ± 7.3 vs. 379.2 ± 7.0, *p* = 0.0001), 10-Trial rats (479.3 ± 10.2 vs. 374.1 ± 6.7, *p* = 0.0001), and all rats (475.7 ± 3.7 vs. 377.9 ± 3.1, *p* = 0.0000; all Wilcoxon). The decline began at 180–200 s and continued for the rest of the 10 min in all groups (Appendix A). It was also observed during the playback sessions (see e.g., Figure 2B,D,F,H) but was never observed during control intervals (Appendix A).

### 3.8. 50-kHz Sounds Caused HR to Increase; 22-kHz Sounds Caused HR to Decrease

Ultrasonic playback affected HR values in all rats (Figure 1, Figure 2 and Figure 3, Appendix A), that is, the analysis of repeated measures revealed crucial differences from 0 s to 30 s time-intervals following 50-kHz playbacks, as well as from −10 s to 10 s time-intervals following 22-kHz playbacks (*p* = 0.0000 in all cases for all rats analyzed together with the exception of 50-kHz tone with *p* = 0.0011, Wilcoxon). These effects were also present in most of the groups analyzed separately, while no effects (*p* > 0.05) were observed during control time-intervals, that is, from −30 s to −10 s or −120 s to −100 s (Appendix A).

The changes in HR around the signal onset, that is, from −10 s to 30 s, were most striking, especially in rats exposed to 50-kHz vs. 22-kHz playback. The former resulted in a significant increase in HR values between 0 s time-interval and following time-intervals (Appendix A). Whereas after 22-kHz sounds presentation, the most striking feature was a drop in HR levels from the intervals before the playback vs. subsequent 10–60 s time-intervals (Appendix A). Moreover, when results following USV and tone playbacks were averaged, that is, when comparing both 50-kHz and both 22-kHz groups of results (Figure 1B,D,F,H; Appendix A), the tendencies of HR levels to increase or decrease, and their significances, intensified.

As a consequence, 50-kHz playback resulted in higher HR following 50-kHz USV- vs. 22-kHz USV playback, 50-kHz tone vs. 22-kHz tone, and especially following averaged 50-kHz sounds vs. 22-kHz sounds in all analyzed groups (Figure 1B,D,F,H). Please note that HR values in response to the 50-kHz vs. 22-kHz sounds differed throughout 0–180 s time-intervals in all the animals combined (Appendix A).

### 3.9. One-Trial Rats Showed Lower HR Levels and a Smaller Response to 50-kHz Playback

The HR of 1-Trial rats was lower than the HR in other groups throughout the experiment. There was a group HR effect for averaged values in the whole initial 10 min interval (Appendix A, *p* = 0.0312, Kruskal-Wallis); 1-Trial rats displayed lower HR (461.0 ± 7.3) when compared not only to control rats (476.6 ± 5.5, *p* = 0.0441) but also to 6-Trial (482.7 ± 7.3, *p* = 0.0135) and 10-Trial groups (479.3 ± 10.2, *p* = 0.0373; all Mann-Whitney).

Similarly, during the playback session (Figure 1B,D,F,H), there was a group effect at the −120 s to −100 s time interval and the entire −120 s to −10 s time interval (*p* = 0.0304, *p* = 0.0436, respectively, Kruskal-Wallis) when data from all presentations, that is, USV, tone, 50-kHz, and 22-kHz, were averaged for analysis. At this latter interval, 1-Trial rats displayed lower HR (386.0 ± 3.5) when compared not only to control rats (402.0 ± 4.6, *p* = 0.0268) but also 6-Trial (405.3 ± 5.3, *p* = 0.0039) and 10-Trial groups (397.3 ± 6.3, not significant *p* = 0.0608; all Mann-Whitney).

In 1-Trial rats, HR response to 50-kHz playback was smaller than in other groups (Figure 1 and Appendix A); this effect was observed especially following 50-kHz USV playback and averaged sound playback (Figure 1D vs. Figure 1B,F,H; Appendix A). For example, the increase in HR between the −10 s time-interval and post playback averaged 10–60 s interval was smaller and even negative in 1-Trial rats following USV playback (−0.4 ± 4.0) and sound playback (−2.0 ± 3.9) vs. HR increase in, for example, 0-Trial rats following USV playback (12.1 ± 3.8, *p* = 0.0254) and 10-Trial rats following sound playback (14.2 ± 5.4, *p* = 0.0341; Mann-Whitney; Appendix A, last row).

### 3.10. Rats Vocalized More Often during and Following 50-kHz Playback Than 22-kHz Playback

The presentation of 50-kHz playback resulted in a dramatic increase in the number of USV emitted (Figure 1, Figure 2 and Appendix A). In contrast, the increase was modest during and after the presentation of the 22-kHz sounds (Figure 1, Figure 2 and Appendix A). When the values of USV emissions following 50- vs. 22-kHz playback were compared (Figure 1B,D,F,H), there was a clear and prolonged difference across all analyzed groups, that is, there were more USV following 50-kHz playback throughout 0–180 s time-intervals (Appendix A).

Moreover, following the analysis of the number and parameters of USV produced by rats from the different groups (Table 1), we observed higher numbers of USV during and following 50-kHz playback, vs. 22-kHz playback, not only regarding all emitted USV (*p* = 0.0000), but also regarding 50-kHz USV (*p* = 0.0000) and short 22-kHz USV (*p* = 0.0017) in particular (see Appendix A for 50- vs. 22-kHz playbacks comparisons). Additionally, 50-kHz USV emitted in response to 50-kHz playback were both longer (27.8 ± 1.0 ms) and of higher frequency (60.5 ± 0.5 kHz) than 50-kHz calls emitted in response to the 22-kHz playback (23.2 ± 1.5 ms, *p* = 0.0000; 57.6 ± 0.9 kHz, *p* = 0.0000; respectively; Appendix A).

### 3.11. Natural and Artificial Ultrasounds Produced Similar Results, but Still, Some Differences Stood Out

In particular, natural 50-kHz playback produced more USV and more 50-kHz USV responses in FC rats (1, 6, and 10-Trial groups), analyzed together (82.3 ± 10.3, 81.0 ± 10.2, respectively), than the 50-kHz tone playback (68.3 ± 9.7, *p* = 0.0283; 67.3 ± 9.7, *p* = 0.0251; respectively, Wilcoxon; Appendix A). This effect was only observed in each FC group but was not significant; it was not observed in the 0-Trial group.

Similarly, natural 22-kHz playback produced more USV and more 50-kHz USV responses in all rats analyzed together (21.4 ± 3.2, 20.6 ± 3.2, respectively) than 22-kHz tone playback (14.2 ± 2.6, *p* = 0.0196; 13.8 ± 2.6, *p* = 0.0238; respectively, Wilcoxon; Appendix A). This effect was observed in most groups but was significant in the 0-Trial group only (*p* = 0.0025, *p* = 0.0018, respectively, Wilcoxon). Additionally, 50- and 22-kHz tone playback resulted in a lower frequency of emitted 50-kHz USV than in case of USV playbacks (Appendix A).

Natural playback evoked a more pronounced approach to the speaker (Appendix A), which was mainly observed at 10–60 s time intervals when results from relevant groups were analyzed together. For all rats, there was a difference in time spent in the speaker’s half of the cage following 50-kHz USV playback (78.0 ± 2.8%) vs. 50-kHz tone playback (67.6 ± 2.9%, *p* = 0.0004), as well as following 22-kHz USV playback (73.1 ± 3.7%) vs. 22-kHz tone playback (60.8 ± 4.0%, *p* = 0.0236; both Wilcoxon; Appendix A).

### 3.12. Previously-Shocked Rats Vocalized More Often, with Longer and Higher Frequency USV

USV of rats from different groups differed in number, duration and frequency (Table 1). There was a group effect for the total number of USV (*p* = 0.0407) and the total number of 50-kHz USV (*p* = 0.0419) emitted to the 50-kHz USV playback. There was also a group effect for the duration of 50-kHz USV emitted during the first 10 min of silence (*p* = 0.0346). The latter effect was also present for averaged responses following 22-kHz playbacks (*p* = 0.0383; Table 1, all Kruskal–Wallis).

The analysis of between-group differences revealed that previously FC rats vocalized more than controls during the whole experiment, 10 min of introductory silence, and following both 50- and 22-kHz playbacks. For example, control rats emitted on average 41.2 ± 7.1 USV during and after 50-kHz USV playback, while all FC rats emitted 82.3 ± 10.3 USV (*p* = 0.0275, Mann–Whitney, Table 1).

Moreover, 50-kHz USV emitted by FC rats were also longer, for example, during the 10 min baseline period (21.6 s ± 1.4 ms vs. 15.7 ± 1.7 ms in control rats; *p* = 0.0025) and following 22-kHz sound playback (28.2 ± 3.4 ms vs. 18.4 ± 1.6 ms in control rats; *p* = 0.0066) and they were also of higher frequency. Mean peak frequency values were higher in five of the six investigated cases of not averaged data (see Table 1), which was significant in the case of USV emitted to the 50-kHz tone playback (*p* = 0.0471).

## 4. Discussion

We used our formerly published model based on rats’ exposure to pre-recorded playbacks in home-cage-like conditions. Several observations were the same as described in our previous publication [26]:−Rats’ overt behavior remained relatively constant except during ultrasonic playback;−Rats moved faster during 50-kHz ultrasonic presentations;−Rats slowed down right after 22-kHz ultrasonic presentations;−Rats approached the speaker during and following 50-kHz and 22-kHz playbacks;−The approach was more pronounced during and following 50-kHz playback;−HR levels declined during the whole experimental session;−Rats’ HR increase after exposure to 50-kHz playback;−Rats’ HR decrease when exposed to 22-kHz playback;−The difference in HR following 50-kHz vs. 22-kHz playback lasts for at least 3 min;−Both 50- and 22-kHz sounds evoked an ultrasonic response, mainly in the 50-kHz range;−50-kHz USV emitted in response to 50-kHz playback were longer and of higher frequency;−Rats vocalized more often during and following 50-kHz playback than 22-kHz playback;−In general, rats reacted in a similar way to both natural and artificial ultrasonic playback.

Rats emitted a high amount of 50-kHz USV in response to our ultrasonic presentation in contrast to other playback studies [29,30,31,32,33], possibly due to the housing-like conditions during recording. We also observed behavioral changes in animals when faced with different playbacks, which included approaching the speaker. It was shown before that playback of 50-kHz calls causes approach behavior in rats, however 22-kHz presentation was repeatedly reported to elicit behavioral inhibition and no-social-approach [33,34,35,36,37,38,39,40] and hiding [41], while we observed speaker-side preference following 22-kHz playback (Figure 1, Appendix A). The preference could again be a result of the low-stress, home-cage-like experimental conditions. Please note that there was still a decrease in locomotor activity following 22-kHz playback [39]. Another reason for the discrepancy could be the difference in playback duration, which was only 10 s in our case, while others have investigated up to 10–15 min (e.g., [41,42]).

Along with visible behavioral alternations, we observed changes in rats’ HR with a striking difference in HR response to 50-kHz and 22-kHz playback. These changes could be explained by the emotional arousal evoked by the two different call types, which activate specific limbic and cortical areas of the brain. These are predominantly the frontal and motor cortices and nucleus accumbens for 50-kHz calls as well as the perirhinal cortex, basolateral amygdala, and periaqueductal gray—for 22-kHz USV [25,29]. The nucleus accumbens is responsible for modulating appetitive behaviors and is regulated by dopaminergic afferent fibers. A sudden increase in HR correlated with an emergent approach behavior following playback of 50-kHz USV [43], while activation of the periaqueductal gray, which is regarded as a defense–response center, following playback of 22-kHz USV was accompanied by reduced locomotor activity and freezing [44,45]. According to the polyvagal theory, physiological changes such as the regulation of HR, respiratory rhythm along with several behaviors, for example, vocal emissions, are intrinsically linked via a common signaling pathway—the vagus nerve [46,47,48]. Therefore sensory stimulation by 50-kHz or 22-kHz USV most likely lead to system-wide physiological changes including cardiovascular, locomotor and vocal reactions.

The electric shock protocol is used to study physiological associative aversive memory and/or to imitate traumatic events leading to pathological conditions. Notably, exposure to even a single foot-shock session was shown to induce long-lasting inhibition of activity in the shock-context and in unknown environments that markedly differ from the shock context. This effect is known as fear generalization [49]. Delivery of a higher number of shocks with higher amperage, frequency and longer shock duration represents a more severe traumatic stressor than a lower number of shorter, lower amperage shocks; see Figure 2 in [3]. In our study, a single 1 mA, 1 s shock served as the weakest stimulus while repeating it six and ten times increased the severity of the treatment. Rats that received six and ten shocks had higher freezing levels than those receiving only one.

To the best of our knowledge, USV emission in reaction to USV playback has never been studied in the context of previously experienced shock. We propose that the observed reactions can broadly be interpreted as a sign of hypervigilance. Our conditioned rats showed higher locomotor activity during the 50-kHz playback and a more significant decrease in activity following the 22-kHz playback. Increased locomotor activity during appetitive playback and decreased activity immediately following the aversive playback were previously observed [26]; however, these reactions are intensified in FC rats (Figure 2A,C,E,G).

Similarly, induced hypervigilance with exaggerated reactions has been previously observed in rat models of PTSD. For example, previously shocked rats showed increased avoidance reactions and unnecessary crossings after cessation of foot-shock [50]. Previously shocked rats buried unfamiliar objects, while control animals did not [51]. Correspondingly, PTSD patients showed physiological and behavioral hyperreactivity to environmental stressors even if they are not related to the traumatic situation, for example, exaggerated acoustic startle responses [52]. Furthermore, increased startle has been reported during experimental induction of fear in healthy individuals, especially in high-fear subjects [53].

In this study, a sign of hypervigilance was more frequent vocalization by previously-shocked rats compared to control animals. Their 50-kHz USV were longer and of higher frequency as well (Figure 2B,D,F,H and Figure 3B; Table 1). Interestingly, a similar finding was observed in single-reared rats that vocalize more often than paired ones to ultrasonic playback [26]. It is worth noting that the rearing of rats in isolation causes both anxiety-like and depression-like symptoms [54,55]. It was hypothesized before that the peak frequency, along with the number of calls per time unit, is involved in coding the quantitative aspect of 50-kHz calls [56]. A recent study reported on a higher frequency of 50-kHz USV appearing in a foot-shock paradigm where one animal in a pair was witnessing the other receiving the aversive stimuli. A fraction of high-frequency (>75 kHz) USV were observed in pairs where the observer animal was naïve to the testing conditions and the foot-shocks were preceded by an audible cue [57]. Finally, the aforementioned dopamine system also modulates the mechanisms underlying fear and anxiety and supports the acquisition of conditioned fear with a potential key role of dopamine receptors in supporting amygdaloid synaptic plasticity underlying the consolidation of the CS–US association [58]. Dopamine receptors’ antagonism was demonstrated to result in reduced call rate, increased latency to call, decreased duration, intensity, bandwidth and peak frequency [59].

The electric shock affected HR levels as well. However, the effects were not linear, that is, they did not correlate with shock intensity. In particular, 1-Trial rats showed lower HR levels and a smaller response to 50-kHz playback in HR increase than control and 6- and 10-Trial rats (Appendix A). The stress-induced changes in HR have been shown before not to be directly proportional to the intensity of the stressor and have been postulated to be a function of stress severity and duration, which translate into a differential stimulation and dynamic balance between sympathetic and parasympathetic (vagal) tones in the heart [12,60,61].

For example, a reduction of HR, which lasted five days, was observed following subchronic fear conditioning. This effect was shown to be predominantly vagally-mediated, that is, the post-stress vagal tone was higher compared with the prestress level [13]. Similarly, FC rats showed less pronounced tachycardia when compared with control animals, which was attributed to simultaneous activation of the sympathetic nervous system and parasympathetic nervous system. In contrast, in non-shocked controls, a predominant sympathetic nervous system activation results in a more significant increase in HR [12].

In humans, vagal tone dominates in healthy resting conditions. In this study, the increase in the vagal tone may cause an HR decrease observed in mildly shocked 1-Trial rats. On the other hand, chronically stressed humans are often characterized by anomalies in the autonomic regulation of HR, such as elevated sympathetic and reduced vagal tone, which can induce tachycardia [13]. The increased HR observed in 6- and 10-Trial rats might model emotional trauma and anxiety observed in humans to correlate with lowering the vagal tone in the heart and increasing the sympathetic tone [60]. PTSD patients were shown to have more significant cardiac responses to startling sounds and idiosyncratic trauma reminders [62,63].

Our results are of limited generalizability, since only male rats were used in the experiments. Estimates from community studies suggest that women are two to three times more likely to develop PTSD than men, while USA prevalence estimates of lifetime PTSD from the National Comorbidity Survey Replication are 9.7% for women and 3.6% for men (https://www.ptsd.va.gov/understand/common/common_adults.asp, assessed on 30 June 2021). Therefore, relevant future research should include female rats.

## 5. Conclusions

Our ultrasonic playback–answer behavioral paradigm in rats combined with the previously applied electric shock can serve as a model of hypervigilance associated with past trauma and PTSD syndrome according to DSM-5 [64]. Therefore, the detection of increased vocalization can serve as a valuable new measure of hypervigilance for the behavioral animal modeling of PTSD. Future pharmacological evaluation of this measurement should be considered.

## Figures and Tables

**Figure 1 brainsci-11-00970-f001:**
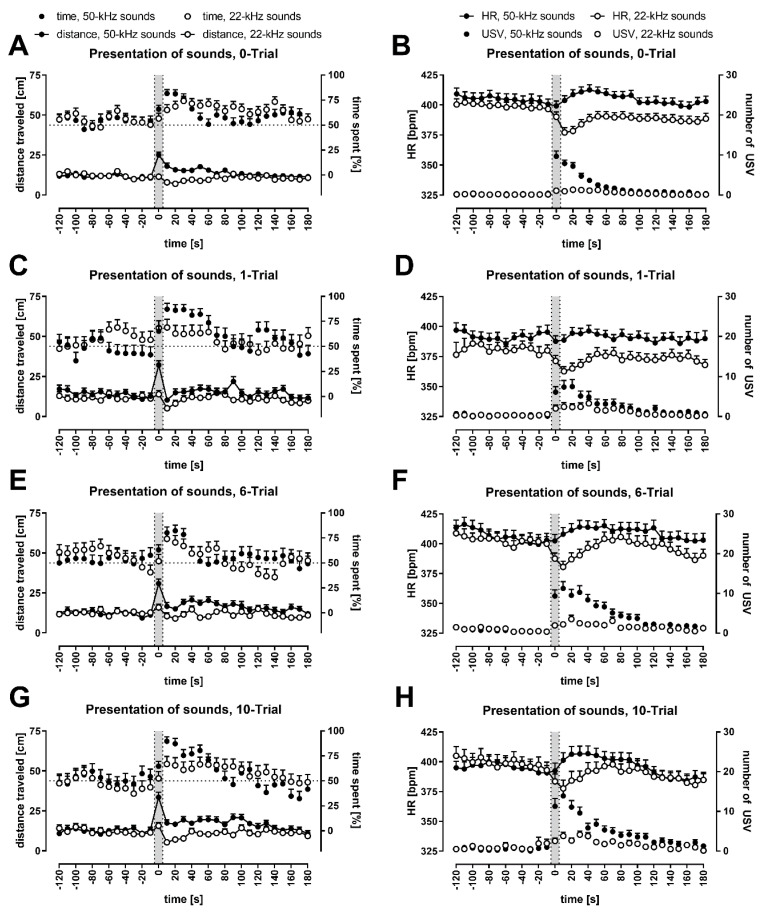
Analysis of changes in distance traveled, time spent in the speaker’s half of the cage, heart rate (HR), and the number of emitted vocalizations during playback session in rats exposed to both 50-kHz- vs. 22-kHz playback, with averaged effects of natural and artificial sounds. Gray sections correspond to the 10-s-long ultrasonic presentation. Graphs depict responses after previous exposure to: no shock (**A**,**B**), one shock (**C**,**D**), six shocks (**E**,**F**), and ten shocks (**G**,**H**). In the left column (**A**,**C**,**E**,**G**), distance traveled is presented as connected dots (cm, left Y axis), percentage of time spent in the speaker’s cage half—as not connected dots (%, right Y axis). In the right column (**B**,**D**,**F**,**H**), HR is presented as connected dots (bpm; beats per minute, left Y axis); the number of USV is presented as not connected dots (right Y axis). Each point is a mean for a 10-s-long time-interval with SEM. The dotted horizontal line marks a 50% chance value for time in a side of the cage. Playback of 50-kHz sounds results in a rise of locomotor activity (the weakest in control rats), copious USV emissions and HR increase (the weakest in 1-Trial rats). Playback of 22-kHz sounds is followed by decrease in locomotor activity (the smallest in 0-Trial rats) and HR as well as modest increase in vocalization; groups: 0-Trial, *n* = 37; 1-Trial, *n* = 16; 6-Trial, *n* = 20; 10-Trial, *n* = 19.

**Figure 2 brainsci-11-00970-f002:**
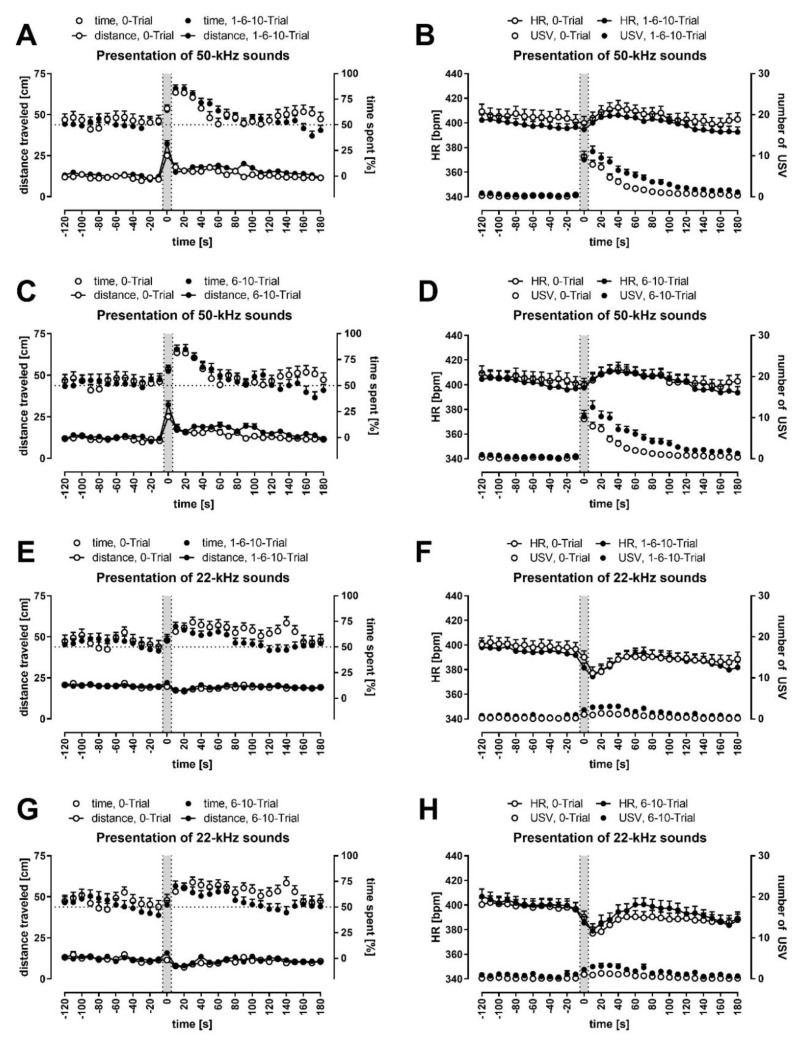
Impact of prior fear conditioning on distance traveled, time spent in the speaker’s half of the cage, heart rate (HR) and USV emission. Gray sections correspond to the 10-s-long ultrasonic presentation. Graphs depict responses to 50-kHz sounds (**A**–**D**) and 22-kHz sounds (**E**–**H**), that is, with averaged effects of natural and artificial sounds in control animals (0-Trial) vs. FC rats (1-6-10-Trial combined, (**A**,**B**,**E**,**F)** 6-10-Trial combined, (**C**,**D**,**G**,**H)**). In the left column (**A**,**C**,**E**,**G**), distance traveled is presented as connected dots (cm, left Y axis), percentage of time spent in the speaker’s cage half—as not connected dots (%, right Y axis). In the right column (**B**,**D**,**F**,**H**), HR is presented as connected dots (bpm; beats per minute, left Y axis); the number of USV is presented as not connected dots (right Y axis). Each point is a mean for a 10-s-long time-interval with SEM. The dotted horizontal line marks a 50% chance value for time in a side of the cage. FC rats had higher locomotor activity during 50-kHz playback, a more significant decrease in activity following 22-kHz playback and more USV in response to playback; groups: 0-Trial, *n* = 37; 1-Trial, *n* = 16; 6-Trial, *n* = 20; 10-Trial, *n* = 19; 1-6-10-Trial, *n* = 55; 6-10-Trial, *n* = 39.

**Figure 3 brainsci-11-00970-f003:**
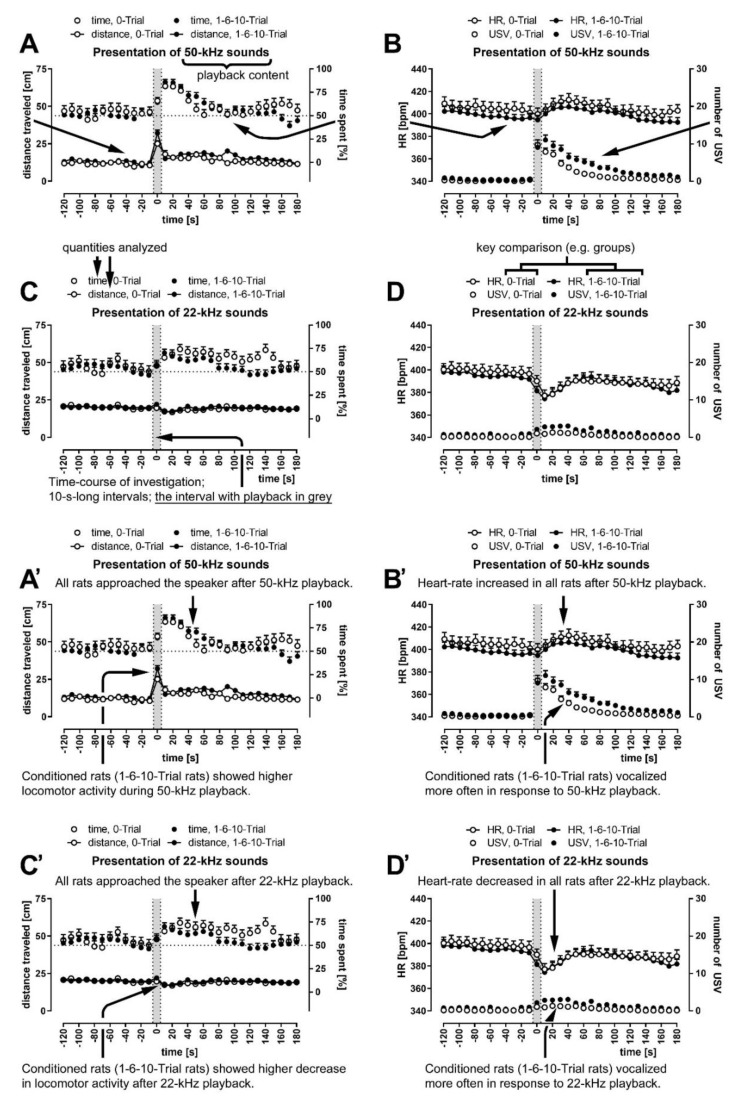
(**A**–**D**) Main results from Figure 2A,B,E,F with guiding explanations (arrows) pointing to measured quantities and crucial graph elements. (**A’**–**D’**) Main take-home messages in the form of short titles connected (arrows) with important data points.

**Table 1 brainsci-11-00970-t001:** Comparison of number of USV of different types and selected characteristics of 50-kHz USV emitted during the whole experiment, first 10 min of silence and during playback sessions, that is, during the 10-s-long playback and 110 s afterwards in response to 50- vs. 22-kHz USV, tone and sounds (averaged results) in control (0-Trial, *n* = 37) and fear-conditioned rats (1-Trial, *n* = 16; 6-Trial, *n* = 20; 10-Trial, *n* = 19; all fear-conditioned/FC’ed). MPF—mean peak frequency; categories of USV: 50-kHz (MPF >32 kHz), short 22-kHz (MPF of 18–32 kHz, duration <0.3 s), long 22-kHz (18–32 kHz, >0.3 s); underlined: effect of group showed by the Kruskal–Wallis test; * *p* < 0.05; ** *p* < 0.01, differences between 0-Trial and other groups; # *p* < 0.05, the difference between 1-Trial vs. 10-Trial groups, all Mann–Whitney test. Please note, USV emitted by FC rats are more numerous, prolonged and, usually, of higher frequency.

Groups	Number of USV				Parameters of 50-kHz USV	
	Total USV	50-kHz USV	Short 22-kHz	Long 22-kHz	Duration [ms]	MPF [kHz]
**USV emitted during the whole experiment**						
0-Trial	152.7 ± 24.2	141.6 ± 23.8	10.7 ± 2.6	0.4 ± 0.4	25.8 ± 1.2	58.0 ± 0.9
1-Trial	230.0 ± 68.0	205.7 ± 66.7	22.2 ± 8.3	2.0 ± 2.0	26.3 ± 2.0	58.3 ± 1.5
6-Trial	366.1 ± 84.9	346.7 ± 85.2	9.1 ± 1.9	10.4 ± 7.1	25.3 ± 2.1	61.7 ± 1.1
10-Trial	354.2 ± 84.7 **	336.1 ± 84.2 **	10.7 ± 2.2	7.3 ± 7.3	29.0 ± 1.8	60.1 ± 0.4
all FC’ed	322.4 ± 46.8 *	302.0 ± 46.7	13.5 ± 2.7	6.9 ± 3.6	26.9 ± 1.1	60.2 ± 0.6
**USV emitted during the first 10 min of silence**						
0-Trial	23.9 ± 7f.1	19.4 ± 7.0	4.5 ± 1.0	0.0 ± 0.0	15.7 ± 1.7	52.1 ± 1.9
1-Trial	53.6 ± 18.0 *	38.4 ± 15.6	13.1 ± 6.6	2.0 ± 2.0	22.2 ± 3.4	52.7 ± 2.9
6-Trial	72.8 ± 26.6	57.8 ± 26.6	4.6 ± 1.2	10.4 ± 7.1	21.7 ± 2.1 *	55.5 ± 1.1
10-Trial	49.7 ± 23.1	38.1 ± 22.3	4.4 ± 1.3	7.3 ± 7.3	21.0 ± 2.0 *	56.6 ± 1.7
all FC’ed	59.2 ± 13.4	45.3 ± 13.0	7.0 ± 2.0	6.9 ± 3.6	21.6 ± 1.4 **	55.0 ± 1.2
**USV emitted to 50-kHz USV playback (0–120 s time-intervals)**						
0-Trial	41.2 ± 7.1	39.9 ± 6.9	0.9 ± 0.2	0.3 ± 0.3	27.6 ± 1.7	60.5 ± 1.5
1-Trial	59.1 ± 19.5	57.8 ± 19.3	1.3 ± 0.6	0.0 ± 0.0	24.8 ± 2.2	60.4 ± 1.1
6-Trial	86.4 ± 16.5	85.7 ± 16.5	0.8 ± 0.3	0.0 ± 0.0	27.3 ± 2.6	62.8 ± 1.4
10-Trial	97.5 ± 17.5 **	95.8 ± 17.2 **	1.7 ± 0.7	0.0 ± 0.0	30.6 ± 1.8	60.9 ± 0.7
all FC’ed	82.3 ± 10.3 *	81.0 ± 10.2 *	1.2 ± 0.3	0.0 ± 0.0	27.8 ± 1.3	61.4 ± 0.6
**USV emitted to 22-kHz USV playback (0–120 s time-intervals)**						
0-Trial	13.5 ± 3.6	12.9 ± 3.6	0.6 ± 0.3	0.0 ± 0.0	21.5 ± 1.9	58.7 ± 1.8
1-Trial	14.6 ± 5.4	13.4 ± 5.3	1.1 ± 0.6	0.0 ± 0.0	31.3 ± 13.1	55.9 ± 3.3
6-Trial	32.2 ± 7.6 *	31.9 ± 7.6	0.3 ± 0.1	0.0 ± 0.0	29.4 ± 2.4 *	60.1 ± 0.7
10-Trial	30.9 ± 9.8	29.9 ± 9.8	1.1 ± 0.5	0.0 ± 0.0	23.8 ± 2.4	58.9 ± 1.8
all FC’ed	26.6 ± 4.7	25.8 ± 4.7	0.8 ± 0.3	0.0 ± 0.0	27.7 ± 3.5	58.5 ± 1.1
**USV emitted to 50-kHz tone playback (0–120 s time-intervals)**						
0-Trial	41.4 ± 6.6	40.4 ± 6.5	1.0 ± 0.4	0.0 ± 0.0	27.5 ± 1.3	59.1 ± 0.7
1-Trial	49.3 ± 16.2	48.6 ± 16.1	0.8 ± 0.2	0.0 ± 0.0	25.4 ± 2.2	60.6 ± 1.4
6-Trial	76.0 ± 18.5	74.6 ± 18.4	1.4 ± 0.5	0.0 ± 0.0	30.1 ± 2.9	60.1 ± 1.1
10-Trial	76.3 ± 15.3	75.4 ± 15.2	0.9 ± 0.4	0.0 ± 0.0	31.7 ± 2.2	60.6 ± 0.6
all FC’ed	68.3 ± 9.7	67.3 ± 9.7	1.0 ± 0.2	0.0 ± 0.0	29.3 ± 1.4	60.5 ± 0.6 *
**USV emitted to 22-kHz tone playback (0–120 s time-intervals)**						
0-Trial	6.6 ± 2.1	6.1 ± 2.0	0.5 ± 0.3	0.0 ± 0.0	17.5 ± 2.1	55.5 ± 1.9
1-Trial	16.0 ± 6.9	15.3 ± 6.9	0.7 ± 0.5	0.0 ± 0.0	27.8 ± 7.1	54.9 ± 2.7
6-Trial	18.1 ± 5.5 *	17.8 ± 5.5	0.4 ± 0.2	0.0 ± 0.0	26.9 ± 3.3 *	60.1 ± 0.9
10-Trial	23.3 ± 8.1	23.2 ± 8.1 *	0.1 ± 0.1	0.0 ± 0.0	25.1 ± 2.2 *	58.3 ± 1.4
all FC’ed	19.3 ± 3.9 *	18.9 ± 3.9 *	0.4 ± 0.2	0.0 ± 0.0	26.5 ± 2.4 **	57.9 ± 1.0
**USV emitted to 50-kHz sound playback (0–120 s time-intervals)**						
0-Trial	41.3 ± 6.0	40.1 ± 5.9	1.0 ± 0.3	0.1 ± 0.1	28.0 ± 1.4	59.9 ± 0.8
1-Trial	54.2 ± 17.4	53.2 ± 17.2	1.0 ± 0.4	0.0 ± 0.0	25.1 ± 1.8	60.5 ± 1.1
6-Trial	81.2 ± 16.4	80.1 ± 16.3	1.1 ± 0.3	0.0 ± 0.0	27.6 ± 2.7	62.4 ± 1.4
10-Trial	86.9 ± 15.1 *	85.6 ± 15.0 **	1.3 ± 0.4	0.0 ± 0.0	31.2 ± 1.9 #	60.8 ± 0.6
all FC’ed	75.3 ± 9.4	74.2 ± 9.3	1.1 ± 0.2	0.0 ± 0.0	28.2 ± 1.3	61.2 ± 0.6
**USV emitted to 22-kHz sound playback (0–120 s time-intervals)**						
0-Trial	10.1 ± 2.7	9.5 ± 2.6	0.6 ± 0.3	0.0 ± 0.0	18.4 ± 1.6	58.3 ± 1.6
1-Trial	15.3 ± 4.7	14.4 ± 4.7	0.9 ± 0.4	0.0 ± 0.0	36.5 ± 11.6	53.7 ± 3.1
6-Trial	25.2 ± 5.9 *	24.8 ± 5.9 *	0.3 ± 0.1	0.0 ± 0.0	26.8 ± 2.4 **	60.4 ± 0.7
10-Trial	27.1 ± 8.3 *	26.6 ± 8.3 *	0.6 ± 0.3	0.0 ± 0.0	23.2 ± 2.2	58.0 ± 1.6
all FC’ed	23.0 ± 3.8 *	22.4 ± 3.8 *	0.6 ± 0.2	0.0 ± 0.0	28.2 ± 3.4 **	57.6 ± 1.1

## Data Availability

Raw data, analyzed herein, have been deposited to Mendeley Data at https://data.mendeley.com/datasets/3pbnnxjzv7/1.

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
