# Peer review of "Increased Vocalization of Rats in Response to Ultrasonic Playback as a Sign of Hypervigilance Following Fear Conditioning"

_brainsci, 2021, doi:10.3390/brainsci11080970_

Round 1

Reviewer 1 Report

The idea of this study has an interesting target. However, some misconceptions and the way the results were shown, mask any interesting aspect of the findings. The title explores the concept of fear conditioning in response to ultrasonic playbacks. To start, this study is not about FC but footshock. Animals have to be successful conditioned to be said that they are part of a fear conditioning experiment. This study did not show that the animals acquired fear (through freezing behaviour, usually measured after the footshock), thus they are not conditioned. No freezing was shown reading the footshock day. Freezing behaviour was evaluated on D2 (a day after the ultrasonic test) and there is no distinction of context and cued apparatus (were they the same? Is the ultrasonic box the same than the footshock box?). Also, there is no indication of how freezing behaviour was evaluated (during the cue only, during the ITI and cue…).

I appreciate the effort to perform 1, 6 and 10 footshock but I believe that a better rationale is needed. Plus, 1, 6 and 10 footshock created plenty of additional graphs that just make the understanding of the data difficult to follow. Because of this, it is also difficult to find where exactly are the statistical differences in the results or some of the data described (I may be blind but could not find the average speed data). I would recommend redesigning the graph to make it clear. There must be a clear reason for the data to be displayed in the main text. I did not understand the importance of speaker side. It looks obvious for that animals will generally go to the speaker side since there is familiar noise there. Also, the choice of distance travelled also need more background. What is the relevance for the study? Should animals increase ambulatory behaviour after a noise? Why not food intake since one of the ultrasonic sounds is associated with appetite?

Other comments:

Surgical procedures and experiment design: I would suggest to the authors to expand this section of the manuscript. In addition, an illustrative image with the surgery details and the experimental design would help the reader.

What was the rational behind the use of 1, 6 or 10 conditioning stimuli instead of focusing on one trial with X stimuli?  Since the authors wish to investigate “fear conditioning”, why not go the common 5X footshock?

Stats: Which method did the authors use to identify the outliers?

Results 3.1: How freezing was measured during the test 2 days after the conditioning? Just during the 20s blocks? Entire time of the test? If they differ, how did the context box differ from the cue box?

Results 3.2: would the possible neophobia be avoid if a mock speaker was presented on the “non-speaker” side?

L434: what is and how is defined a “physiological stimulus”? This is not term commonly used in Fear conditioning studies.

Discussion must be improved to be less descriptive.

Author Response

Response to comments of Reviewer 1

  1. The title explores the concept of fear conditioning in response to ultrasonic playbacks. To start, this study is not about FC but footshock.

We understand that the previous wording of the title may suggest that the ultrasonic playback was a part of the fear conditioning itself, like CS, therefore, we have changed the title to: “Increased vocalization of rats in response to ultrasonic playback as a sign of hypervigilance following fear conditioning”.

  1. Animals have to be successful conditioned to be said that they are part of a fear conditioning experiment. This study did not show that the animals acquired fear (through freezing behaviour, usually measured after the footshock), thus they are not conditioned. No freezing was shown reading the footshock day. Freezing behaviour was evaluated on D2 (a day after the ultrasonic test)

In our opinion, the rats were successfully fear-conditioned as they acquired fear-response reaction, observed through freezing (Fig. S1C). The reaction could be observed immediately after conditioning but can also be measured at different time-points following conditioning. We would not agree with the prerequisite to show freezing on the very day of conditioning for allowing the term “fear conditioning”. We are not exactly sure whether we understand precisely Reviewer’s criticism concerning this point. Additionally, however, we would argue and state that fear conditioned rats were conditioned on day 0 and stayed conditioned on day 1, during the key experiment, which is proven by the freezing test done on day 2.

Evaluation of freezing following fear-conditioning is rarely done the same day or during conditioning session (Doenni et al., 2017); usually it is performed the following day (Lindquist et al., 2004; Detert et al., 2008; Burman et al., 2010; Likhtik et al., 2014; Madarasz et al., 2016) or during two consecutive days following conditioning-day (Wohr et al., 2005; Esclassan et al., 2009; Botterill et al., 2014), or later, e.g. three days (Kong et al., 2014), five weeks (Manassero et al., 2018).

                Finally, our approach is in line with definitions of fear conditioning, e.g. “fear conditioning is a simple form of associative learning, in which an animal learns to associate the presence of a neutral stimulus, termed the conditioned stimulus (CS), such as a light or a tone, with the presence of a motivationally significant stimulus, termed the unconditioned stimulus (US), such as an electric shock to the foot” (Engelmann and Hein, 2013).

  1. There is no distinction of context and cued apparatus (were they the same)?

Yes, there was no distinction. Freezing to context and cue was performed in the same apparatus, i.e. in the same cage. Reaction to the cue (white light) was observed on the top of the reaction to context. Some changes were introduced to highlight this element of methodological approach, e.g. menu and description of Fig. S1.

  1. Is the ultrasonic box the same than the footshock box?

No. These were two different boxes. Fear-conditioning was done in Med Associates apparatus, whereas ultrasonic playback experiment was done in home-like cages (“individual experimental cages, identical to home cages; page 3, line 91)

We have changed the wording in methods (page 3, line 87) to further underline the difference:
“A playback experiment was given one day later (see the next paragraph); the following day (two days after conditioning), rats were returned to the same fear-conditioning context to measure freezing levels (Test).”

  1. Also, there is no indication of how freezing behaviour was evaluated (during the cue only, during the ITI and cue…).

We have added the sentence: “Freezing was evaluated during the habituation and exposure to CS.” Also, the relevant information is in the Fig. S1 legend.

  1. I appreciate the effort to perform 1, 6 and 10 footshock but I believe that a better rationale is needed.

It is hard to come up with more substantial and real rationale for the particular numbers of shocks. As stated, we wanted to investigate the impact of varying numbers of shocks, i.e. “dose-dependency”, not really knowing beforehand which values (number of foot-shocks, amperage) to choose from. Finally, it turned out that some of the effects were observed in 1-Trial group only, while 6- and 10-Trial rats behaved in similar ways. The difficulty comes from the lack of broadly-accepted standards of the “varying numbers of shocks”. However, we have added the sentence about “encompassing usually used range (Aliczki & Haller)” of the number of shocks.

  1. Plus, 1, 6 and 10 footshock created plenty of additional graphs that just make the understanding of the data difficult to follow. Because of this, it is also difficult to find where exactly are the statistical differences in the results or some of the data described (I may be blind but could not find the average speed data).

We added an explanation: “(Please note, graphs contain distances travelled in cm per 10 s; speed, when mentioned, is reflected by the distances divided by 10, i.e. in cm/s)” (page 4, line 142). Also, we have limited the number of graphs.

  1. I would recommend redesigning the graph to make it clear.

We have attempted to make the graphs clearer and more readable by introducing one set of them with explanations, i.e. please see Fig. 3. We would rather not reconstruct the graphs in a more profound way; it took a lot of effort to come up with the current version and we have not been able to propose a better one. Also, the graphs have been designed and published in this particular way before (Olszynski et al., 2020). We are afraid that introducing a different system would definitely confuse readers of both publications.

  1. There must be a clear reason for the data to be displayed in the main text. I did not understand the importance of speaker side. It looks obvious for that animals will generally go to the speaker side since there is familiar noise there.

We have explained the importance of this finding in Discussion.

Our findings do seem important, since they bring somehow different results from the most of observed by others. Possibly an increase in side preference, we observe, happens because rats are kept in low-stress, home-cage-like conditions. Please note, there is still a decrease in locomotor activity following 22-kHz playback.

The problem has been studied in the laboratory of Marcus Wohr (Wohr and Schwarting, 2007; Seffer et al., 2014) that playback of 50-kHz calls causes approach behavior in rats while 22-kHz presentation elicits behavioral inhibition. Differential exploratory behavior influenced by playback of ultrasounds from the 50-kHz and 22-kHz bandwidth was shown in a recent study by (Inagaki and Ushida, 2021). In their experiment they put a rat in a modified open field test with an added concealed hiding spot and a speaker emitting USV recordings. Animals spend significantly more time concealed and decreased exploratory behaviors during 22-kHz and 25-kHz sounds playback when compared to a control 1-kHz long sine wave tone.

  1. Also, the choice of distance travelled also need more background. What is the relevance for the study? Should animals increase ambulatory behaviour after a noise?

Rats have been shown previously to increase locomotor activity during 50-kHz playback. We wanted to see this reaction in previously-shocked rats. It turned out conditioned rats showed higher locomotor activity during playback. Also, rats have been shown previously to decrease locomotor activity after 22-kHz playback; we are showing that the effect is bigger in previously-shocked animals (comp. Fig. 1A vs. CEG).

  1. Why not food intake since one of the ultrasonic sounds is associated with appetite?

We understand the possible confusion concerning the use of the word “appetitive” which usually means pertaining to appetite. However, in the context of USV the other meaning relating to “desirable” and as an opposite of “aversive” 22-kHz is commonly used; 50-kHz ultrasonic vocalizations are regarded as signaling appetitive emotional states. From our unpublished experience, we would argue that when exposed to ultrasonic playback, rats would rather focus on the playback not on food intake.

  1. Surgical procedures and experiment design: I would suggest to the authors to expand this section of the manuscript. In addition, an illustrative image with the surgery details and the experimental design would help the reader.

The section was expanded; we added:

“Radiotelemetric transmitter (HD-S10, Data Sciences International, St. Paul, MN, USA) for cardiovascular studies was disinfected using Cidex® (Johnson&Johnson, New Brunswick, NJ, USA) and implanted under ketamine-xylazine anesthesia. The abdominal region was shaved and disinfected (Octenisept, Schulke, Norderstedt, Germany). A midline incision was performed, the transmitter sensor was implanted into the abdominal aorta by direct puncturing of the vessel (21g needle) and fixed with tissue glue (Histoacryl®, B. Braun, Melsungen, Germany). The transmitter body was placed in the peritoneal cavity and fixed to the abdominal muscle wall. After surgery, the animal was subcutaneously injected with Metacam (0.4 mg/kg; Boehringer Ingelheim, Ingelheim am Rhein, Germany) for analgesia.”

Also, the reader is given an opportunity to consult relevant illustrative image of the surgery details published elsewhere. Towards this end a sentence was added: “Illustrative image with the surgery details can be found elsewhere (Fig. 5 in (Pestana-Oliveira et al., 2020); please note, we used tissue glue instead of cellulose patches and silk sutures.“

  1. What was the rationale behind the use of 1, 6 or 10 conditioning stimuli instead of focusing on one trial with X stimuli?  Since the authors wish to investigate “fear conditioning”, why not go the common 5X footshock?

We have actually used one trial with 1, 6 or 10 shocks; with rationale mentioned above. We would argue there is no pervasive protocol for fear conditioning with set number of foot-shocks. We had verified the relevant literature and found broad range of foot-shocks used: e.g. one (Lindquist et al., 2004), four (Doenni et al., 2017), five (Esclassan et al., 2009), six (Wohr et al., 2005; Carnevali et al., 2011), seven (Botterill et al., 2014), eight (Souza et al., 2018), ten (Detert et al., 2008), and twelve (Burman et al., 2010). Also, our final findings proved our decision to use different numbers of shocks to be right – some results were characteristic for the rats which received one shock; other results were different and characteristic for 6- and 10-Trial animals.

  1. Which method did the authors use to identify the outliers?

In general, we would rather like to refrain from eliminating outliers. However, two rats in 6-Trial group turned to be exceptionally high-vocalizers before conditioning. Since we are showing that 6-Trial rats vocalize more often when exposed to playback, we decided not to include these animals in the relevant group. We eliminated them as “emitting exceptionally many USV, >3 x Standard Deviation” (page 4, line 131). Please note, all significant values reported were also calculated and confirmed with the outliers present (as mentioned in Methods, line 132). The results presented in the original Fig. S1 (Submission 1) did not contain the two rats.

  1. Results 3.1: How freezing was measured during the test 2 days after the conditioning? Just during the 20s blocks? Entire time of the test? If they differ, how did the context box differ from the cue box?

All the questions have been answered above.

  1. Results 3.2: would the possible neophobia be avoid if a mock speaker was presented on the “non-speaker” side?

Yes, it is possible. Such a mock-speaker is used by others, however not always (Wohr and Schwarting, 2007, 2009) or here: (Inagaki and Ushida, 2021). However, we do not finally consider this result to be important and as encouraged by the Reviewers to limit the Result section, we have decided not to mention it.

  1. L434: what is and how is defined a “physiological stimulus”? This is not term commonly used in Fear conditioning studies.

This term was used in the literature comparing foot-shock use to model learning vs. PTSD in rats. We changed “physiological stimulus” to “the weakest stimulus”.

  1. Discussion must be improved to be less descriptive.

We have changed Discussion. Added were: the discussion of the results similar to those obtained before with the special emphasis on the time spent in the speaker’s half of the cage, observed changes in peak frequency, and lack of female rats in our experiments.

Literature cited

Botterill JJ, Fournier NM, Guskjolen AJ, Lussier AL, Marks WN, Kalynchuk LE (2014) Amygdala kindling disrupts trace and delay fear conditioning with parallel changes in Fos protein expression throughout the limbic brain. Neuroscience 265:158-171.

Burman MA, Hamilton KL, Gewirtz JC (2010) Role of corticosterone in trace and delay conditioned fear-potentiated startle in rats. Behav Neurosci 124:294-299.

Carnevali L, Bondarenko E, Sgoifo A, Walker FR, Head GA, Lukoshkova EV, Day TA, Nalivaiko E (2011) Metyrapone and fluoxetine suppress enduring behavioral but not cardiac effects of subchronic stress in rats. Am J Physiol Regul Integr Comp Physiol 301:R1123-1131.

Detert JA, Kampa ND, Moyer JR, Jr. (2008) Differential effects of training intertrial interval on acquisition of trace and long-delay fear conditioning in rats. Behav Neurosci 122:1318-1327.

Doenni VM, Song CM, Hill MN, Pittman QJ (2017) Early-life inflammation with LPS delays fear extinction in adult rodents. Brain Behav Immun 63:176-185.

Engelmann JB, Hein G (2013) Contextual and social influences on valuation and choice. Prog Brain Res 202:215-237.

Esclassan F, Coutureau E, Di Scala G, Marchand AR (2009) Differential contribution of dorsal and ventral hippocampus to trace and delay fear conditioning. Hippocampus 19:33-44.

Inagaki H, Ushida T (2021) The effect of playback of 22-kHz and 50-kHz ultrasonic vocalizations on rat behaviors assessed with a modified open-field test. Physiol Behav 229:113251.

Kong E, Monje FJ, Hirsch J, Pollak DD (2014) Learning not to fear: neural correlates of learned safety. Neuropsychopharmacology 39:515-527.

Likhtik E, Stujenske JM, Topiwala MA, Harris AZ, Gordon JA (2014) Prefrontal entrainment of amygdala activity signals safety in learned fear and innate anxiety. Nat Neurosci 17:106-113.

Lindquist DH, Jarrard LE, Brown TH (2004) Perirhinal cortex supports delay fear conditioning to rat ultrasonic social signals. J Neurosci 24:3610-3617.

Madarasz TJ, Diaz-Mataix L, Akhand O, Ycu EA, LeDoux JE, Johansen JP (2016) Evaluation of ambiguous associations in the amygdala by learning the structure of the environment. Nat Neurosci 19:965-972.

Manassero E, Renna A, Milano L, Sacchetti B (2018) Lateral and Basal Amygdala Account for Opposite Behavioral Responses during the Long-Term Expression of Fearful Memories. Sci Rep 8:518.

Olszynski KH, Polowy R, Malz M, Boguszewski PM, Filipkowski RK (2020) Playback of Alarm and Appetitive Calls Differentially Impacts Vocal, Heart-Rate, and Motor Response in Rats. iScience 23:101577.

Pestana-Oliveira N, Nahey DB, Johnson T, Collister JP (2020) Development of the Deoxycorticosterone Acetate (DOCA)-salt Hypertensive Rat Model. Bio Protoc 10:e3708.

Seffer D, Schwarting RK, Wohr M (2014) Pro-social ultrasonic communication in rats: insights from playback studies. J Neurosci Methods 234:73-81.

Souza RR, Robertson NM, Pruitt DT, Noble L, Meyers EC, Gonzales PA, Bleker NP, Carey HL, Hays SA, Kilgard MP, McIntyre CK, Rennaker RL (2018) The M-Maze task: An automated method for studying fear memory in rats exposed to protracted aversive conditioning. J Neurosci Methods 298:54-65.

Wohr M, Schwarting RK (2007) Ultrasonic communication in rats: can playback of 50-kHz calls induce approach behavior? PLoS One 2:e1365.

Wohr M, Schwarting RK (2009) Ultrasonic communication in rats: effects of morphine and naloxone on vocal and behavioral responses to playback of 50-kHz vocalizations. Pharmacol Biochem Behav 94:285-295.

Wohr M, Borta A, Schwarting RK (2005) Overt behavior and ultrasonic vocalization in a fear conditioning paradigm: a dose-response study in the rat. Neurobiol Learn Mem 84:228-240.

Reviewer 2 Report

The article entitled “Increased Vocalization Following Fear Conditioning in Response to Ultrasonic Playback in Rats” is overall an important contribution to the field. It presents a great amount of data both reproducing previous results (behavioral and HR responses to USV playback) and investigating a novel question (impact of previous FC on such responses). However, the presentation of the data is suboptimal (in particular difficult to follow and to extract the take-home message) and several points would need reworking to improve readability and statistical rigor.  I present them below in no particular order.

  • The statistical comparisons of the data are currently not ideal: the authors need to test if there is a difference in a specific measure overall between groups (tones and USVs ; -, 1-, 6-, and 10-trial groups ) before everything. If there is, then they can analyze them separately. If there is not, then they can group them (as "sounds", as “1-6-10”…) but cannot analyze them separately. This is an either-or situation and they cannot do both. This is challenging to do in non-parametric tests. However, I am not clear as to what the reasoning was to use non-parametric tests vs parametric ones. If the different measures presented are following the prerequisites for parametric tests, then multiple-way ANOVAs with repeated and non-repeated variables would be the test to run before everything to check whether groups should be analyzed separately or together.
  • The figures in general are challenging to follow, and in particular to identify which graph represents what. It took me several re-reading of the legend to figure out the difference between 4A and 4E for example. In general, I feel that there is too much data in each figure. The article readability may benefit from putting some parts of the figures in the supplementary material.
  • It is currently rather difficult to go back and forth to understand which points in the figures are different from each other. It may be helpful if p<0.05 was reported on the figures, if the authors can find a way to make it appear without crowding the figures.
  • With the amount of data presented, what is novel is a little lost in what is reproducing previous findings. Reproducing previous findings is tremendously important and I am in no way suggesting that the authors should take it out. However, could they find a way to condense that part a bit to expend a bit more the novelty?
  • Avoid bar graphs in figures, especially in figure S1 because these data are used to exclude animals. At minimum add scatterplots of individual data (see https://doi.org/10.1371/journal.pbio.1002128)
  • The following points need clarification in the methods:
    • The authors indicate that FC was conducted partly during the light phase partly during the dark phase (FC: 1500 to 2400, light 0900 to 2100) Can they check whether they observe a difference or not in the FC obtained whether the conditioning was conducted during the light or dark phase?
    • The surgical procedure needs to be at least briefly developed in the methods, the sole reference to a previous article is not enough here. At minimum, specify whether aseptic methods where used, the type of anesthesia used, and if antibiotic and analgesics were given.
    • The methods state “An experienced user counted the number of USV manually” but the results describe analyses of the USV frequencies and durations, with a distinction between 22-kHz USV and 50-kHz USV. Please develop the methods used and the software.
  • The authors studied only male rats. Responses to FC can vary between males and females and patients with PTSD are 2/3 female (https://www.ptsd.va.gov/understand/common/common_adults.asp). Therefore, this choice is confusing, if not unfortunate, and deserves to be explained. A note in the discussion could be also added to this point and to the potential generalizability or non-generalizability of the authors’ results to females argued based on existing literature.
  • Overall, a stronger comparison to previous results would benefit the discussion. The authors compare extensively their current results to their previous publication. However, some seminal articles with comparable setups are lacking in the citations here, such as
    • Brudzynski, S.M.; Chiu, E. Behavioural Responses of Laboratory Rats to Playback of 22 KHz Ultrasonic Calls. Physiology & behavior 1995, 57, 1039–1044;
    • Seffer, D.; Schwarting, R.K.W.; Wöhr, M. Pro-Social Ultrasonic Communication in Rats: Insights from Playback Studies. Journal of Neuroscience Methods 2014, 234, 73–81, doi:10.1016/j.jneumeth.2014.01.023;
    • Wöhr, M.; Schwarting, R.K.W. Ultrasonic Communication in Rats: Can Playback of 50-KHz Calls Induce Approach Behavior? PLoS ONE 2007, 2, e1365, doi:10.1371/journal.pone.0001365

to cite only a few.

  • “Rats communicate via several sensory channels, e.g. by emitting ultrasonic vocalizations (USV).” Line 49: replace “g.” by “including”
  • “We recently discovered changes in locomotion, USV emission, and HR in Wistar rats exposed to ultrasonic playback from a speaker as a new social communication model” line 55 could suggest that the changes in locomotion, USV emission and HR are a mean for the rat to communicate. Communication requires an emitter and a receptor, i.e. that a receptor (another rat) could read these changes and interpret their meaning. Here, these changes are rather a response to the reception of a social signal. Please rephrase that sentence.

Author Response

Response to comments of Reviewer 2

  1. However, the presentation of the data is suboptimal (in particular difficult to follow and to extract the take-home message).

We prepared Fig. 3 explaining our way of data presentation and extracted some take-home messages.

  1. The statistical comparisons of the data are currently not ideal: the authors need to test if there is a difference in a specific measure overall between groups (tones and USVs ; -, 1-, 6-, and 10-trial groups) before everything. If there is, then they can analyze them separately. If there is not, then they can group them (as "sounds", as “1-6-10”…) but cannot analyze them separately. This is an either-or situation and they cannot do both. This is challenging to do in non-parametric tests. However, I am not clear as to what the reasoning was to use non-parametric tests vs parametric ones. If the different measures presented are following the prerequisites for parametric tests, then multiple-way ANOVAs with repeated and non-repeated variables would be the test to run before everything to check whether groups should be analyzed separately or together.

We have followed Reviewer’s advice, which resulted in reducing the number of several comparisons described in chapters 3.4, 3.5, and 3.9. – in comparison with the previous version of the Ms. Parametric tests could not be used, since the data violates assumptions for parametric ANOVA – also after extensive and numerous data-transformations tested.

  1. The figures in general are challenging to follow, and in particular to identify which graph represents what. It took me several re-reading of the legend to figure out the difference between 4A and 4E for example. In general, I feel that there is too much data in each figure. The article readability may benefit from putting some parts of the figures in the supplementary material.

We propose to easy the reading of the figures by introducing explanatory Fig. 3. Also, we have agreed and decided to place Figs 1 and 2 (form Submission 1) in the supplementary materials. We would not like to generally change the appearance of the figures, since they are analogous to the ones already published in our previous article (Olszynski et al., 2020).

  1. It is currently rather difficult to go back and forth to understand which points in the figures are different from each other. It may be helpful if p<0.05 was reported on the figures, if the authors can find a way to make it appear without crowding the figures.

We have tried and tested various versions of the figures – by adding different p-values. As it was mentioned before, the figures are challenging to read. Unfortunately, adding the described and/or key p-values (which could now be found in the tables) would absolutely unable the reader to profit from interpretation of the figures.

  1. With the amount of data presented, what is novel is a little lost in what is reproducing previous findings. Reproducing previous findings is tremendously important and I am in no way suggesting that the authors should take it out. However, could they find a way to condense that part a bit to expend a bit more the novelty?

We have condensed and shortened parts of the Ms containing previous findings. In particular Figs 1 and 2 were moved to Supplementary Materials, the mention of neophobia was deleted, several chapters were shortened due to elimination of several redundant comparisons (according to #2).

  1. Avoid bar graphs in figures, especially in figure S1 because these data are used to exclude animals. At minimum add scatterplots of individual data (see https://doi.org/10.1371/journal.pbio.1002128)

Fig. S1 was changed to scatterplot form. Please note, the previous version of Fig. S1 did not contain the outliers. Introducing scatterplots to the other graphs would unable the reader to follow the course of changes in analyzed quantities before, during and after playback; such a modification would entail the introduction of significantly more details to already complicated figures.

  1. The authors indicate that FC was conducted partly during the light phase partly during the dark phase (FC: 1500 to 2400, light 0900 to 2100) Can they check whether they observe a difference or not in the FC obtained whether the conditioning was conducted during the light or dark phase?

We have verified it. There was no effect of the phase on fear-conditioning results.

  1. The surgical procedure needs to be at least briefly developed in the methods, the sole reference to a previous article is not enough here. At minimum, specify whether aseptic methods where used, the type of anesthesia used, and if antibiotic and analgesics were given.

We have developed this section; it now says:

“Radiotelemetric transmitter (HD-S10, Data Sciences International, St. Paul, MN, USA) for cardiovascular studies was disinfected using Cidex® (Johnson&Johnson, New Brunswick, NJ, USA) and implanted under ketamine-xylazine anesthesia. The abdominal region was shaved and disinfected (Octenisept, Schulke, Norderstedt, Germany). A midline incision was performed, the transmitter sensor was implanted into the abdominal aorta by direct puncturing of the vessel (21g needle) and fixed with tissue glue (Histoacryl®, B. Braun, Melsungen, Germany). The transmitter body was placed in the peritoneal cavity and fixed to the abdominal muscle wall. After surgery, the animal was subcutaneously injected with Metacam (0.4 mg/kg; Boehringer Ingelheim, Ingelheim am Rhein, Germany) for analgesia. Illustrative image with the surgery details can be found elsewhere (Fig. 5 in (Pestana-Oliveira et al., 2020); please note, we used tissue glue instead of cellulose patches and silk sutures.”

  1. The methods state “An experienced user counted the number of USV manually” but the results describe analyses of the USV frequencies and durations, with a distinction between 22-kHz USV and 50-kHz USV. Please develop the methods used and the software.

We have added the section: “USV recordings were analyzed using SASLab Pro 5.2.xx. Spectrograms were generated from the .wav files with the following parameters: window type: FlatTop, 512 FFT length, 100% frame size and 75% temporal resolution overlap. An experienced user scored USV on the spectrogram. For analysis, mean peak frequency (MPF) and element duration were taken via values measured by the software.”

  1. The authors studied only male rats. Responses to FC can vary between males and females and patients with PTSD are 2/3 female

(https://www.ptsd.va.gov/understand/common/common_adults.asp). Therefore, this choice is confusing, if not unfortunate, and deserves to be explained. A note in the discussion could be also added to this point and to the potential generalizability or non-generalizability of the authors’ results to females argued based on existing literature.

A note in the discussion was added: “Our results are of limited generalizability, since only male rats were used in the experiments. Estimates from community studies suggest that women experience PTSD at two to three times the rate that men do while U.S. prevalence estimates of lifetime PTSD from the National Comorbidity Survey Replication are 9.7% for women and 3.6% for men (https://www. ptsd.va.gov/understand/common/common_adults.asp). Therefore, the relevant future research should include female rats.”

  1. Overall, a stronger comparison to previous results would benefit the discussion. The authors compare extensively their current results to their previous publication. However, some seminal articles with comparable setups are lacking in the citations here, such as

Brudzynski, S.M.; Chiu, E. Behavioural Responses of Laboratory Rats to Playback of 22 KHz Ultrasonic Calls. Physiology & behavior 1995, 57, 1039–1044;

Seffer, D.; Schwarting, R.K.W.; Wöhr, M. Pro-Social Ultrasonic Communication in Rats: Insights from Playback Studies. Journal of Neuroscience Methods 2014, 234, 73–81, doi:10.1016/j.jneumeth. 2014.01.023;

Wöhr, M.; Schwarting, R.K.W. Ultrasonic Communication in Rats: Can Playback of 50-KHz Calls Induce Approach Behavior? PLoS ONE 2007, 2, e1365, doi:10.1371/journal.pone.0001365

to cite only a few.

We have included a stronger comparison to previous results. The discussion was expanded. The references have been included.

  1. “Rats communicate via several sensory channels, e.g. by emitting ultrasonic vocalizations (USV).” Line 49: replace “g.” by “including”

The text was corrected.

  1. “We recently discovered changes in locomotion, USV emission, and HR in Wistar rats exposed to ultrasonic playback from a speaker as a new social communication model” line 55 could suggest that the changes in locomotion, USV emission and HR are a mean for the rat to communicate. Communication requires an emitter and a receptor, i.e. that a receptor (another rat) could read these changes and interpret their meaning. Here, these changes are rather a response to the reception of a social signal. Please rephrase that sentence.

The sentence was rephrased; it was shortened.

Literature cited

Olszynski KH, Polowy R, Malz M, Boguszewski PM, Filipkowski RK (2020) Playback of Alarm and Appetitive Calls Differentially Impacts Vocal, Heart-Rate, and Motor Response in Rats. iScience 23:101577.

Pestana-Oliveira N, Nahey DB, Johnson T, Collister JP (2020) Development of the Deoxycorticosterone Acetate (DOCA)-salt Hypertensive Rat Model. Bio Protoc 10:e3708.

Round 2

Reviewer 1 Report

Happy to see the improvement of the manuscript!